# Integrative Analysis of Differentially Expressed miRNAs and Noncoding RNA Networks Reveals Molecular Mechanisms Underlying Metritis in Postpartum Dairy Cows

**DOI:** 10.3390/cimb47080643

**Published:** 2025-08-11

**Authors:** Ramanathan Kasimanickam, Joao Ferreira, Vanmathy Kasimanickam

**Affiliations:** 1College of Veterinary Medicine, Washington State University, Pullman, WA 99164, USA; vkasiman@wsu.edu; 2School of Veterinary Medicine and Animal Science, São Paulo State University—UNESP, Botucatu 18618-81, SP, Brazil; joao.cp.ferreira@unesp.br

**Keywords:** postpartum cows, uterus, inflammation, immune function, blood, microRNAs, mRNAs, in silico analysis

## Abstract

Postpartum metritis in dairy cows compromises reproductive performance and leads to substantial economic losses. This study investigated the molecular mechanisms underlying metritis by integrating high-throughput circulating microRNA (miRNA) profiling with systems-level bioinformatics. Previously, 30 differentially expressed miRNAs, 16 upregulated and 14 downregulated, were identified in metritis-affected cows compared to healthy controls. Building on these findings, this study predicted miRNA target genes and constructed regulatory networks involving miRNAs, mRNAs, circRNAs, lncRNAs, and snRNAs, alongside protein–protein interaction networks. Functional annotation and KEGG pathway analysis revealed that upregulated miRNAs influenced genes involved in immune activation, apoptosis, and metabolism, while downregulated miRNAs were associated with angiogenesis, immune suppression, and tissue repair. Hub genes such as AKT3, VEGFA, and HIF1A were central to immune and angiogenic signaling, whereas UBE3A and ZEB1 were linked to immune inhibition. Interferon-stimulated genes (e.g., ISG15, RSAD2, CXCL chemokines) were shown to regulate solute carriers, contributing to immune dysregulation. Key pathways included PI3K-Akt, NF-κB, JAK-STAT, insulin resistance, and T cell receptor signaling. Noncoding RNAs such as NEAT1, KCNQ1OT1, and XIST, along with miRNAs like bta-miR-15b and bta-miR-148a, emerged as pro-inflammatory regulators, while bta-miR-199a-3p appeared to exert immunosuppressive effects. These findings offer new insights into the complex regulatory networks driving metritis and suggest potential targets for improving fertility in dairy cows.

## 1. Introduction

Postpartum uterine diseases are categorized as puerperal metritis, clinical metritis, clinical endometritis, and subclinical endometritis based on the time of occurrence after calving and presence of clinical or subclinical signs [1,2,3,4]. Puerperal metritis was defined by an enlarged, flaccid uterus with a foul smelling, watery red-brown discharge, along with pyrexia and systemic illness occurring within 10 days of calving [4]. Other clinical signs include anorexia, depression, and decreased milk yield and feed intake [4,5,6].

In general, postpartum uterine diseases, irrespective of the type, inflict financial losses on dairy farms, mostly due to treatment costs and disposal of milk, decreased milk production, poor reproductive performance, and increased culling [2,5,6]. Although reproductive performance is compromised in dairy cows experiencing postpartum uterine disease, the exact mechanism by which the fertility is affected is not fully understood.

Gene expression of key inflammatory cytokines in circulation varied between normal cows versus cows experiencing uterine disease [7,8]. Similarly, gene expression of cytokines also differed in an inflamed versus normal endometrium of postpartum cows [9,10]. We recently reported the influence of uterine inflammation negatively affecting the length of gestational day 16 conceptus [11]. Although several studies have considered genetic components of uterine inflammation [12,13,14,15], few have elucidated epigenetic changes such as altered expression of regulatory RNAs and their integration with coding genes involved in bovine metritis. The potential regulatory role of microRNAs (miRNAs) in the development and progression of bovine subclinical endometritis has been investigated by studying expression of miRNAs in uterine endometrial samples [13,14].

Recent studies have highlighted the role of noncoding RNAs (ncRNAs), including long noncoding RNAs (lncRNAs), circular RNAs (circRNAs), and small noncoding RNAs (sncRNAs), in the regulation of immune responses, inflammation, and tissue repair [15]. The altered expression of these ncRNAs has been associated with disrupted cellular processes in the uterus [16,17]. Understanding how specific ncRNAs are dysregulated during uterine disease provides valuable insight into the molecular mechanisms underlying infertility and may guide the development of diagnostic or therapeutic strategies.

In our previous study, 84 circulating miRNAs were profiled in postpartum dairy cows, identifying 30 differentially expressed (DE) miRNAs, 16 upregulated and 14 downregulated, in cows with metritis compared to healthy controls [18]. Building on these findings, the present study employed integrative bioinformatics analyses to predict target genes, protein–protein interactions, and regulatory networks involving miRNAs, circRNAs, lncRNAs, snRNAs, and mRNAs associated with metritis.

## 2. Materials and Methods

MicroRNA data from our previous study [18] were used, in which Holstein dairy cows with metritis (*n* = 4) and healthy controls (*n* = 4) were selected. Blood samples were collected via coccygeal venipuncture for serum miRNA profiling using RT-PCR. For clarity, we have included a brief description of the methodology employed. Briefly, serum samples were processed, small RNAs were extracted and reverse transcribed, and mature miRNA expression was profiled using the Qiagen miScript PCR array, which targets 84 bovine miRNAs. Normalization was performed using cel-miR-39-3p and the global CT mean. Data analysis included CT quality control, normalization, and calculation of ΔCT values, fold changes, and statistical significance using a web-based tool.

In the current study we employed integrative bioinformatics analyses to predict target genes, protein–protein interactions, and regulatory networks involving miRNAs, circRNAs, lncRNAs, snRNAs, and mRNAs associated with metritis, which are presented below.

### 2.1. Conserved Nucleotide Sequences

Nucleotide sequences of DE-miRNAs were retrieved from miRBase, (www.mirbase.org accessed on 10 January 2025) and compared for sequence conservation between human and cattle [19,20]. Bovine sequences were very similar to human nucleotide sequences. Therefore, human miRNA IDs were used to construct miRNA-–mRNA interaction network and functional enrichment analysis.

### 2.2. Prediction and Analysis of Target Genes of Differentially Expressed miRNAs

The target genes of DE-miRNAs were predicted using miRNet (http://www.mirnet.ca/ accessed on 10 January 2025) [21]. This tool integrated data from multiple miR databases including TarBase, miRTarBase, and miRecords. The target prediction analysis was performed separately for upregulated and downregulated DE-miRNAs.

### 2.3. Construction of Protein–Protein Interaction Network and Screening of Hub Gene

The protein–protein interaction (PPI) network for predicted target genes of DE-miRNAs’ was generated the Search Tool for the Retrieval of Interacting Genes/Proteins (STRING) online database (http://string-db.org/ accessed on 10 January 2025) [22]. Gene Ontology (GO) functional annotations and Kyoto Encyclopedia of Genes and Genomes (KEGGs) pathway enrichment analyses were conducted on these predicted targets. A *p*-value < 0.05 was regarded as statistically significant. The PPI network was exported to Cytoscape (version 3.9, accessed on 10 January 2025) for visualization [23]. Hub genes were identified as the top 30 nodes in the PPI network using the Maximal Clique Centrality (MCC) method [24], which is known for high precision in identifying essential proteins. Further analysis was conducted using ClueGO [25] (accessed on 10 January 2025) to integrate GO terms and KEGG pathways, generating functionally organized term networks (k score = 3). This tool supports the analysis of single or multiple gene lists and provides comprehensive visualization of functionally related terms.

### 2.4. Gene Ontology and Functional Annotation Analysis

To explore the biological significance of DE-miRNAs and their associated genes, biological processes were analyzed using the PANTHER (Protein ANalysis THrough Evolutionary Relationships) Classification System (https://pantherdb.org accessed on 13 January 2025). Additional GO terms and their co-occurring terms were investigated using QuickGO (https://www.ebi.ac.uk/QuickGO accessed on 13 January 2025).

Further analysis of hub genes included their roles, human tissue expression, and protein–protein interactions (up to 6 closely related genes) using data from STRING (http://string-db.org/ accessed on 13 January 2025) and the human protein atlas (https://www.proteinatlas.org accessed on 13 January 2025). Interferon-stimulated genes associated with up- and downregulated miRNAs in cows with metritis were extracted from miRNet’s target prediction output (http://www.mirnet.ca/ accessed on 13 January 2025).

### 2.5. miRNA, circRNA, lncRNA, snRNA, and mRNA Interaction Network

Interaction networks among miRNA, circRNA, lncRNA, snRNA, and mRNA were generated using miRNet (http://www.mirnet.ca/ accessed on 29 May 2025), separately for upregulated and downregulated miRNAs. Specific interaction maps, including miRNA–circRNA–mRNA, miRNA–lncRNA–mRNA, and miRNA–snRNA–mRNA networks, were also constructed.

## 3. Results

### 3.1. miRNA, Gene, and Protein–Protein Interactions

Of 84 prioritized miRNA previously profiled, 30 miRNAs were differentially expressed (*p* < 0.05; fold regulation ≥ 2), 16 upregulated and 14 downregulated (Figure 1), in circulation in cows with metritis compared to normal cows. These DE-miRNAs were analyzed to predict their target genes. Among the 16 upregulated miRNAs, 10 predicted 74 genes (Appendix A), while 10 of the 14 downregulated miRNAs predicted 123 genes (Appendix A).

Figure 2A presents PPI network for target genes of upregulated miRNAS (72 nodes; 281 edges; PPI enrichment *p* < 1.0 × 10^−16^) revealing 511 significantly (False Discovery Rate (FDR) *p* < 0.05) enriched GO biological processes and 77 significantly (FDR *p* < 0.05) enriched KEGG pathways (Appendix A). Figure 2B shows PPI network for the downregulated miRNAs (120 nodes and 189 edges, PPI enrichment *p* < 1.11 × 10^−14^) with 299 significantly (FDR *p* < 0.05) enriched GO biological processes and 41 significant (FDR *p* < 0.05) KEGG pathways (Appendix A).

PPI networks were constructed using the STRING database and visualized in Cytoscape. To interpret the functionally nested ontology and pathway annotation networks for genes targeted by up- and downregulated DE-miRNAs in cows with metritis, ClueGO analyses were performed and visualized in Figure 3A–C and Figure 4A–C, respectively.

The top 30 hub genes identified using Maximal Clique Centrality (MCC) method for upregulated and downregulated DE-miRNAs are presented in Figure 5A and Figure 5B, respectively. Functional annotation and nested network analysis of these hub genes via ClueGO are shown in Figure 6 and Figure 7. Table 1A,B list the hub genes, their roles, and human tissue expression PPI (up to six closely related genes) for up- and downregulated miRNAs. Table 2 details interferon-stimulated genes targeted by DE-miRNAs in cows with metritis.

### 3.2. Interaction Network Among miRNA, circRNA, lncRNA, snRNA, and mRNA

Network analysis of the 16 upregulated miRNAs, including circRNA, lncRNA, snRNA, and mRNA interactions, revealed connections with 6490 unique circRNAs, 135 snRNAs, 357 lncRNAs, and 2169 genes (Appendix A). The full interaction network is illustrated in Figure 8. In addition, specific miRNA to RNA interactions are detailed in Figure 9A (miRNA to circRNA), Figure 9B (miRNA to lncRNA), and Figure 9C (miRNA to snRNA).

Similarly, analysis of the 14 downregulated miRNAs revealed interactions with 6313 unique circRNAs, 144 snRNAs, 337 lncRNAs, and 3401 genes (Appendix A). The network is shown in Figure 10. In addition, miRNA to circRNA, miRNA to lncRNA, and miRNA to snRNA interactions are shown in Figure 11A, Figure 11B, and Figure 11C, respectively.

Table 3 summarizes the top five coding and noncoding RNAs, ranked by high degree and betweenness centrality, targeted by the DE-miRNAs along with their endometrial expressions and potential functional roles.

## 4. Discussion

Recent advances in high-throughput techniques have enabled the representation of experimental data as networks, where nodes (proteins, transcripts, or metabolites) are connected by edges to illustrate interactions among them. Network analysis helps to elucidate the role of individual proteins and their interactions, the protein–protein interactions. Centrality, a network ranking of biological components, has been widely used to identify influential nodes in complex biological networks [26]. These observed nodes with higher degrees (i.e., more connections) are more likely to represent essential proteins that significantly influence the biological process.

Using these network-based methods, key biological mechanisms contributing to metritis and reduced reproductive performance in dairy cows were identified. In general, abnormal metabolic profile during the periparturient period interferes with immune function and predisposes the cow to postpartum uterine disease and subsequent infertility. Analysis of upregulated miRNAs and their associated genes in cows with metritis revealed a wide range of predicted biological processes both positively and negatively. Key predicted processes included regulation of cellular activities (metabolic process, biosynthetic process, protein modification, multicellular organismal), cell death, proliferation, stress responses, mitosis, cell cycle regulation, cell cycle arrest, adhesion, communication, endogenous stimuli, differentiation, maturation, motility, angiogenesis, receptor signaling, intracellular signal transduction, cellular component organization, development, and response to hypoxia, lipids, vitamins and ions, and hormone stimuli. In addition, processes such as immune response-regulating receptor signaling and regulation of T cells and leukocytes were also implicated. These biological processes are critical for timely uterine involution and any disruptions in these biological processes may result in delayed and impaired uterine recovery after calving.

Interestingly, differentially upregulated miRNAs were associated with the cellular response to vitamins A (GO:0033189 and GO:0071300) and E (GO:0071306). In cows with metritis, higher lipid peroxidation and lower plasma concentrations of vitamin A and vitamin E were observed during the first 3 weeks postpartum compared to healthy cows [27]. Beta-carotene, a precursor to vitamin A, is essential for cellular health. Improved reproductive efficiency has been associated with increased postpartum β-carotene concentration in plasma [28]. Although dietary β-carotene supplementation during the dry period did not affect ovarian activity, progesterone production, or the diameters of the cervix and uterine horns [29], its association with hydroxyproline, a key component in uterine tissue repair, suggests that it may help reduce uterine inflammation and accelerate uterine involution after calving.

In contrast, prepartum supplementation with Selenium (Se), alpha-tocopherol, or both, did not improve postpartum ovarian activity, uterine involution, or reduce clinical abnormalities [30]. However, vitamin E and Se supplementation decreased the incidence of metritis, reduced the number of services per conception, and shortened days open, though they had no effect on retained fetal membranes [31]. Notably, prepartum Se treatment, administered 3 weeks before calving, significantly hastened uterine involution in cows diagnosed with metritis [32]. Collectively, vitamins A and E appear to reduce the incidence of uterine diseases, promote uterine recovery, and support earlier resumption of ovarian cyclicity in postpartum dairy cows.

Additional biological processes predicted from DE-miRNAs included cellular response to lipids (GO:0033993), regulation of ovarian follicle development (GO:0001541) embryo development (GO:0009790, GO:0043009), and embryo implantation (GO:0007566). The positive effects of fatty acid supplementation around calving on reproductive performance in dairy cows have been well documented [33,34].

For example, dairy cows supplemented with calcium salts of safflower oil (SO) from 30 days prepartum to 35 days postpartum exhibited improved innate immunity. Neutrophil functions, phagocytosis, and oxidative burst at 4 days postpartum (dpp), and oxidative burst at 7 dpp, were significantly higher in SO-fed cows compared to those fed palm oil (PO). Neutrophil surface expression of L-selectin and cytokine production (TNF-α, IL-1β) at 35 dpp were also increased in SO-fed cows [34]. These immunomodulatory effects may enhance the cow’s ability to combat bacterial challenges during the postpartum period.

Furthermore, supplementation with unsaturated fatty acids (UFAs) from flaxseed reduced pregnancy losses [35]. Diets enriched with UFAs from flaxseed or sunflower sources promoted embryo development [36], while linolenic acid supplementation led to the development of significantly larger ovarian follicles [35]. Additionally, long-chain fatty acid supplementation postpartum was shown to hasten the resumption of ovarian cyclicity and reduce the incidence of cystic ovarian degeneration [37]. These findings indicate that lipid supplementation not only supports immune function but also improves uterine health, follicular development, and embryo viability.

KEGG pathway analysis of genes associated with upregulated miRNAs revealed enrichment in several key signaling pathways, including PI3K-Akt, relaxin, p53, AMPK/MAPK, ErbB, JAK-STAT, IGF receptor, T cell receptor, HIF-1, TNF, FoxO, estrogen, GnRH, NF-κB, C-type lectin receptor, and Ras pathways. These networks involve genes and their products that mediate cellular responses to intrinsic and extrinsic stimuli. They regulate essential processes such as DNA replication, chromosome segregation, cell division, metabolism, proliferation, survival, growth, and angiogenesis, processes that are critically involved in postpartum uterine recovery and reproductive function.

The PI3K-Akt signaling pathway is a key intracellular signal transduction pathway that regulates metabolism, cell proliferation, survival, growth, and angiogenesis in response to extracellular stimuli. Dysregulation of this pathway has been implicated in endometriosis in women, where elevated levels of PI3K and phosphorylated AKT (Ser473) contribute to abnormal cell proliferation [38,39,40]. It should be noted that hormones could play an essential role in these regulatory pathways; for instance, estradiol (E2) promotes endometriotic cell proliferation through the reduced expression of PTEN and the subsequent activation of AKT [41].

In cows, pathogenic *Escherichia coli* lipopolysaccharide induces endometrial inflammation via the Toll-Like Receptor (TLR)4-IRAK-TRAF4-NF-κB signaling axis. Activation of interleukin-1 receptor-associated kinase (IRAK) and TNF receptor-associated factor 4 (TRAF4) by LPS leads to nuclear translocation of NF-κB, triggering inflammation and apoptosis in bovine endometrial cells [42].

Analysis of upregulated miRNAs and their associated genes in metritis cows also predicted enrichment of insulin receptor binding (GO:0005158), insulin resistance (KEGG pathway), and insulin-like growth factor receptor (IGFR) signaling (GO:0048009). Insulin plays a critical role in homeorhesis during the transition period. Both insulin secretion and tissue responsiveness to insulin are altered postpartum. A disrupted metabolic profile—specifically impaired PMN function—has been linked to increased susceptibility to postpartum uterine diseases and infertility [43]. Cows that developed uterine disease had lower circulating glucose and reduced glycogen concentrations in their PMNs [41]. Reduced glycogen storage likely impairs the oxidative burst capacity of PMNs, weakening immune defense mechanisms and predisposing cows to uterine infections [44].

### Key miRNAs and Their Functional Roles

The top five highly upregulated miRNAs in cows with metritis were bta-miR-15b, bta-miR-17-3p, bta-miR-16b, bta-miR-148a, and bta-miR-26b. MicroRNAs are involved in the regulation of several biological functions and pathways. The proposed functions of miR-15b include hypoxia, angiogenesis, and apoptosis; miR-16 is associated with hypoxia, angiogenesis, inflammation, cell growth, and apoptosis; miR-26 is involved in cell proliferation, myogenesis, cell cycle progression, BMP signaling, and cell differentiation; miR-142 is linked to immune suppression. Increased expression of miR-15b reduced activities of anti-apoptotic protein BCL2 [45], whereas miR-145 inhibits endometriotic cell proliferation, invasiveness, and stemness through regulation of cytoskeletal elements, cell adhesion molecules, and proteolytic factors [46]. Notably, bta-miR-26b, derived from intrauterine extracellular vesicles, contributes to suppression of maternal neutrophil-mediated immunity, thereby possibly promoting uterine inflammation [47]. Despite the generally pro-inflammatory profile of the upregulated miRNAs, miR-148a appears to have a compensatory anti-inflammatory role, countering uterine inflammation through the TLR4–NF-κB axis [48].

In contrast, the top five downregulated miRNAs were bta-miR-148b-5p, bta-miR-199a-3p, bta-miR-122-5p, bta-miR-200b-3p, and bta-miR-10a-5p in cows with metritis. miR-148b target genes TNFRSF10B are involved in regulation of T cell function and apoptosis and enhanced bactericidal activity [49]. MicroRNA-199a-3p suppresses high glucose-induced inflammation by regulating the IKKβ/NF-κB signaling pathway in epithelial cells [50]. Further, miR-199a-3p is significantly lower in women with endometriosis compared to normal women [51]. MicroRNA-200 family regulates cellular proliferation and migration in the endometrium, and downregulation may interfere with regulation of the postpartum immune response in women [52,53]. The downregulation of these miRNAs likely impairs immune function, cellular repair, and inflammatory resolution, potentially contributing to persistent uterine inflammation and delayed involution.

GO/pathway terms specific for upregulated hub genes included regulation of transcription involved in G1/S transition of mitotic cell cycle to prevent mitotic catastrophe, beta-catenin binding, phosphatidylinositol phosphate kinase activity, filopodium, and negative regulation of vasculature development and poly-purine tract binding. Regulatory mechanisms including crosstalk between TLR and Wnt/β-catenin signaling pathways may elicit both pro- and anti-inflammatory functions that show involvement in both normal and pathological conditions [54]. Phosphatidylinositol phosphate kinase is essential for the activation of the signaling pathways regulating cytokine production, cell cycle progression, survival, T cell metabolism, and focal adhesions [55,56]. Filopodia have roles in sensing, migration, and cell–cell interaction. In macrophages, filopodia act as phagocytic tentacles and pull bound objects towards the cell for phagocytosis [57]. Both innate and adaptive immune cells are involved in the mechanisms of endothelial cell proliferation, migration, and activation. Anti-angiogenic cytokines such as IFN-gamma and IL-12 [58] may be involved in negative regulation of vasculature development and tissue repair in the uterus of postpartum cows.

GO/pathway terms specific for downregulated hub genes were angiogenesis, endoderm formation, morphogenesis, regulation of cell differentiation, macrophage differentiation, and regulation of cyclin-dependent protein serine/threonine kinase activity, essential to drive cell cycle progression and transition into different phases [59]. Chemokine family CXL ligand (CXCL) mRNAs are upregulated in dairy cows with endometritis [9,10]. Further, upregulated expression of bta-miR-101 may be associated with TLR2. The overexpression of TLR2 is associated with the activation of the NF-κB-mitogen-activated protein kinase B (MAP3KB) complex and consequent upregulation of pro-inflammatory cytokines and chemokines [60]. This overexpression of these inflammatory cytokines and chemokines can induce severe endometrial tissue damage postpartum and adversely prolong uterine involution and negatively impact future pregnancy outcomes. We observed that interferon-stimulated genes were downregulated in cows with subclinical endometritis, which may adversely affect embryo elongation [9,10]. Differentially expressed (DE) miRNAs and their associated ISG targets are listed in Table 2.

Insights from cancer biology have expanded our understanding of the role of noncoding RNAs in uterine diseases of dairy cows. Among upregulated lncRNAs, KCNQ1OT1 and NEAT1 are of particular interest. KCNQ1OT1 is involved in epigenetic regulation and may influence immune-related gene expression [61,62]. NEAT1 is essential for paraspeckle formation and amplifies inflammatory signaling [63]. Its overexpression in uterine tissue has been linked to chronic inflammation and impaired healing. The expression of TUG1 was significantly decreased in HUVECS following lipopolysaccharide treatment in a time-dependent manner [64]. Other lncRNAs such as XIST, HELLPAR, and TUG1 are also linked to immune modulation and epithelial cell function; their dysregulation may contribute to uterine dysfunction [65,66,67,68,69].

Upregulated circRNAs such as RANBP2, RGPD4, and NBPF10 may influence intracellular transport and immune signaling, potentially exacerbating inflammation [70,71,72,73,74]. Meanwhile, sncRNAs including SNORD17 regulate RNA modification and protein synthesis, with elevated levels indicating cellular stress responses [75,76].

Conversely, the downregulation of lncRNAs like SNHG16, HCG18, and NEAT1 may impair immune responses and placental development [77,78,79,80]. Similarly, decreased expression of circRNAs (KPNA6, MACF1) and sncRNAs (SNORA66) may disrupt in cytoskeletal organization and RNA processing [81,82,83].

It should be noted that some lncRNAs such as KCNQ1OT1, NEAT1, and XIST appear to have dual roles. KCNQ1OT1 affects gene expression and several cellular functions including cell proliferation, migration, epithelial–mesenchymal transition (EMT), apoptosis, viability, autophagy, and inflammation [84]. NEAT1 functions as an miRNA sponge, regulating gene expression involved in cell growth, invasion, EMT, stemness, and resistance to chemotherapy and radiotherapy [85]. •XIST, while originally known for X-chromosome inactivation, also influences immune gene expression and participates in multiple signaling pathways, including TGF-β, PI3K/AKT, Wnt/β-catenin, FOXO, NF-κB, mTOR, MAPK, Toll-like receptor, JAK-STAT, and T/B cell receptor pathways [86,87]. Upregulation of XIST may lead to aberrant gene silencing and immune dysregulation, whereas its downregulation may impair chromosomal regulation and immune homeostasis, worsening uterine dysfunction.

In our network analysis, KCNQ1OT1 exhibited both upregulation and downregulation in different contexts, suggesting its role in immune homeostasis. When upregulated, it may suppress immune-related gene expression and prolong inflammation. When downregulated, it may interfere with genomic imprinting and immune regulation, hindering uterine recovery and increasing infection susceptibility. NEAT1, essential for paraspeckle formation, modulates inflammatory gene expression. Its upregulation may enhance innate immune responses but also cause chronic inflammation and tissue damage. Conversely, NEAT1 downregulation may impair protective immunity and delay healing—both conditions negatively affecting uterine health and fertility. XIST is key to X-chromosome inactivation but also regulates immune gene expression. Its upregulation may cause abnormal gene silencing and immune imbalance, while downregulation can impair chromosomal function and immune regulation, further aggravating uterine disorders.

We observed that the regulatory roles of noncoding RNAs vary widely across biological processes. However, these RNAs often exert their effects by interacting with miRNAs to modulate downstream gene expression. Although the role of noncoding RNAs in many signaling pathways remains underexplored, their functional significance is becoming increasingly evident. Continued research into their mechanisms will likely uncover new insights and therapeutic targets in uterine disease.

## 5. Conclusions

This study used high-throughput miRNA profiling and integrative network analyses to explore molecular mechanisms underlying postpartum uterine disease, particularly metritis in dairy cows. Differentially expressed miRNAs and their target genes were implicated in key biological processes and signaling pathways involved in immune regulation, inflammation, metabolism, cellular homeostasis, and reproduction.

Protein–protein interaction and ClueGO analyses linked these miRNAs to numerous enriched GO terms and KEGG pathways, highlighting roles in cellular stress, angiogenesis, immune signaling, and hormonal regulation—critical for uterine involution and reproductive recovery. Centrality analyses identified hub genes associated with inflammation resolution and tissue remodeling.

Interaction networks involving miRNAs and noncoding RNAs (circRNAs, lncRNAs, snRNAs) revealed complex regulatory mechanisms. Notably, NEAT1, XIST, and KCNQ1OT1 exhibited dual roles in modulating inflammatory responses, potentially influencing disease progression or recovery.

The study also identified miRNA-regulated pathways tied to vitamins A and E, lipid metabolism, insulin resistance, and immune function, factors linked to postpartum health. These findings suggest that metritis involves widespread dysregulation of immune and metabolic pathways, possibly worsened by nutritional deficiencies.

In summary, this integrative transcriptomic analysis provides a detailed molecular view of postpartum uterine health, highlighting miRNA, and noncoding RNA networks plays a role in disease development and recovery, offering promising biomarkers and therapeutic targets for improving fertility in dairy cows.

## Figures and Tables

**Figure 1 cimb-47-00643-f001:**
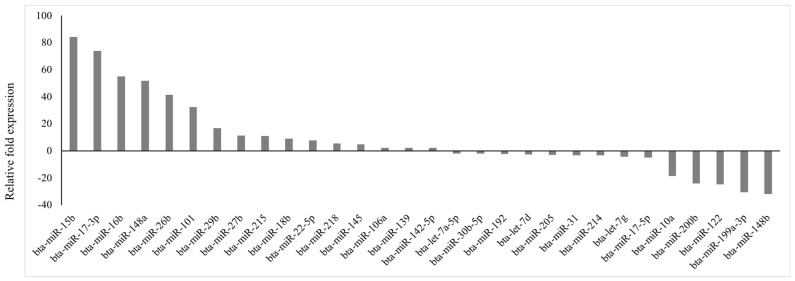
Fold regulation of differentially expressed miRNAs in Holstein cows with metritis of 84 bovine-specific well-characterized miRNAs investigated, 16 were upregulated (*p* ≤ 0.05; fold ≥ 2) and 14 were downregulated (*p* ≤ 0.05; fold ≤ −2) in circulation.

**Figure 2 cimb-47-00643-f002:**
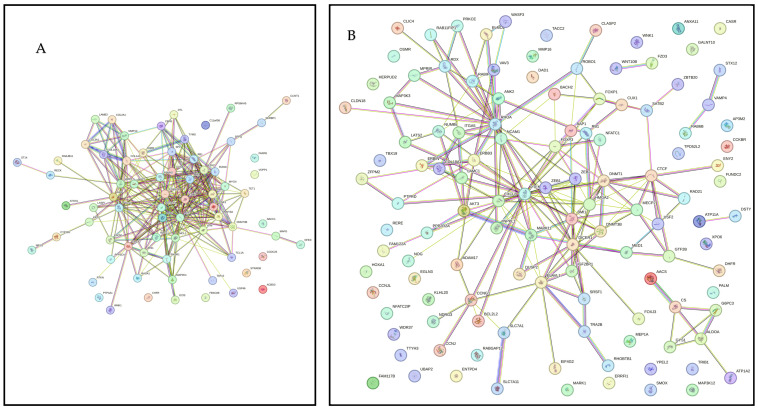
STRING protein–protein interaction (PPI) network. (**A**) PPI network for the upregulated miRNAS-predicted 74 genes (72 nodes and 281 edges, PPI enrichment *p* < 1.0 × 10^−16^). (**B**) PPI network for the downregulated miRNAS-predicted 123 genes (120 nodes and 189 edges, PPI enrichment *p* < 1.11 × 10^−14^).

**Figure 3 cimb-47-00643-f003:**
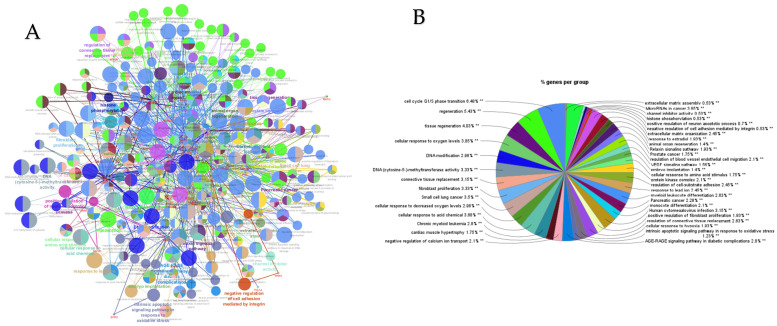
ClueGO analysis of upregulated genes in circulation in cows with metritis. (**A**) Functionally grouped network with terms as nodes linked based on their kappa score level (≥0.4), where only the label of the most significant term per group is shown. The node size represents the term enrichment significance. Functionally related groups partially overlap. The color gradient shows the gene proportion of each cluster associated with the term. (**B**) Overview chart with functional groups including specific terms for upregulated genes. The percentage of genes per term is shown as term label. ** *p* < 0.001; (**C**) GO/pathway terms specific for upregulated genes. The percentage of genes per term is shown as bar label.

**Figure 4 cimb-47-00643-f004:**
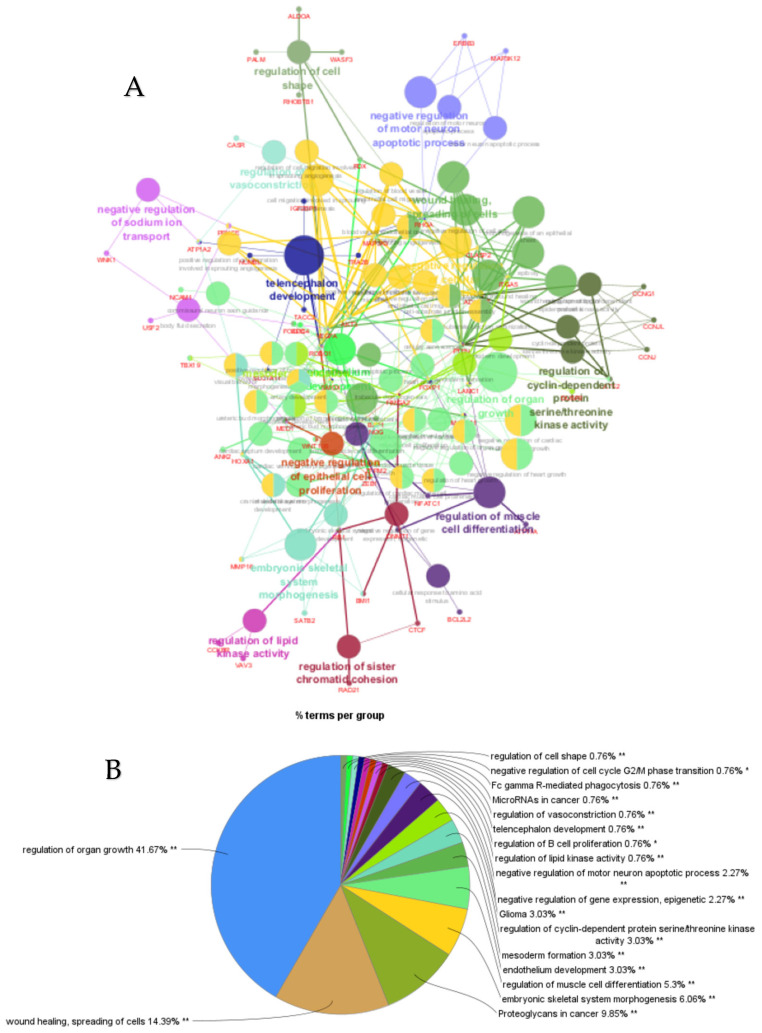
ClueGO analysis of downregulated genes in circulation in cows with metritis. (**A**) Functionally grouped network with terms as nodes linked based on their kappa score level (≥0.4), where only the label of the most significant term per group is shown. The node size represents the term enrichment significance. The color gradient shows the gene proportion of each cluster associated with the term. (**B**) Overview chart with functional groups including specific terms for downregulated genes. The percentage of genes per term is shown as term label. ** *p* < 0.001, * *p* < 0.01. (**C**) GO/pathway terms specific for downregulated genes. The percentage of genes per term is shown as bar label.

**Figure 5 cimb-47-00643-f005:**
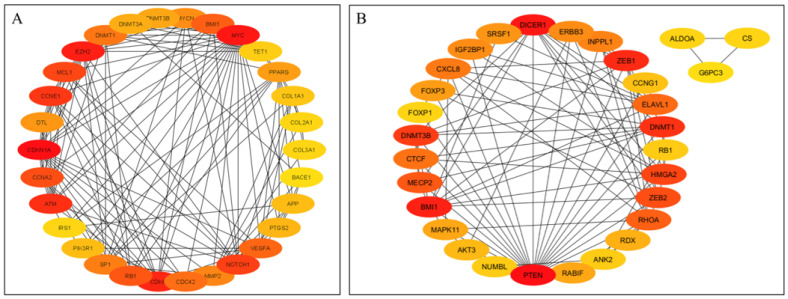
Protein–protein interaction (PPI) network of hub genes of DE-miRNAs. (**A**) PPI network of top genes for highly upregulated DE-miRNAs. (**B**) PPI network of the top genes for downregulated DE-miRNAs. DE-miRNAs differentially expressed microRNAs; black lines indicate interactions between genes. The PPIs among hub genes for upregulated DE-miRNAs were greater compared with hub genes for downregulated DE-miRNAs. A complete list of hub genes are provided in Table 1.

**Figure 6 cimb-47-00643-f006:**
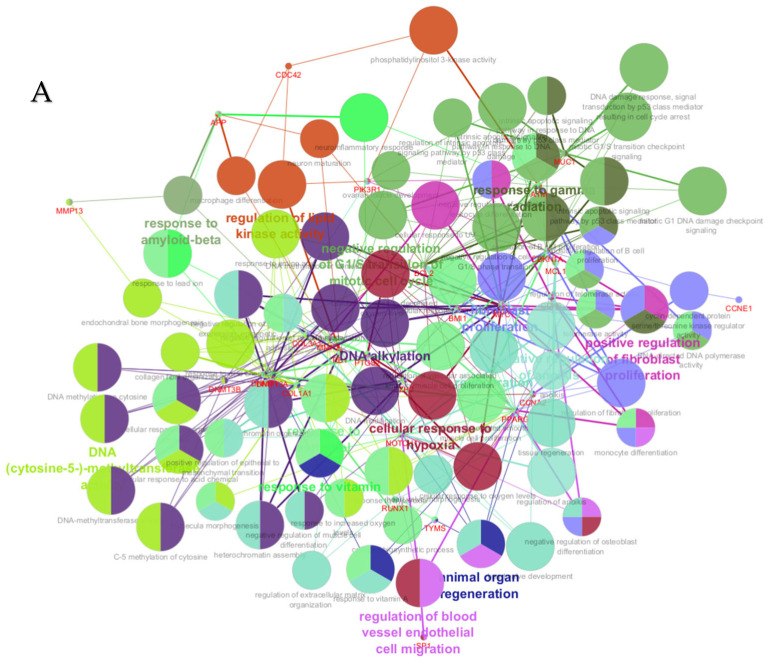
ClueGO analysis of upregulated hub genes in circulation in cows with metritis. (**A**) GO/pathway terms specific for upregulated hub genes. The bars represent the number of genes associated with the terms. The number of genes per term is shown as bar label. (**B**) Functionally grouped network with terms as nodes linked based on their kappa score level (≥0.4), where only the label of the most significant term per group is shown. The node size represents the term enrichment significance. The color gradient shows the gene proportion of each cluster associated with the term. (**C**) Overview chart with functional groups including specific terms for upregulated genes. The percentage of genes per term is shown as term label. ** *p* < 0.001.

**Figure 7 cimb-47-00643-f007:**
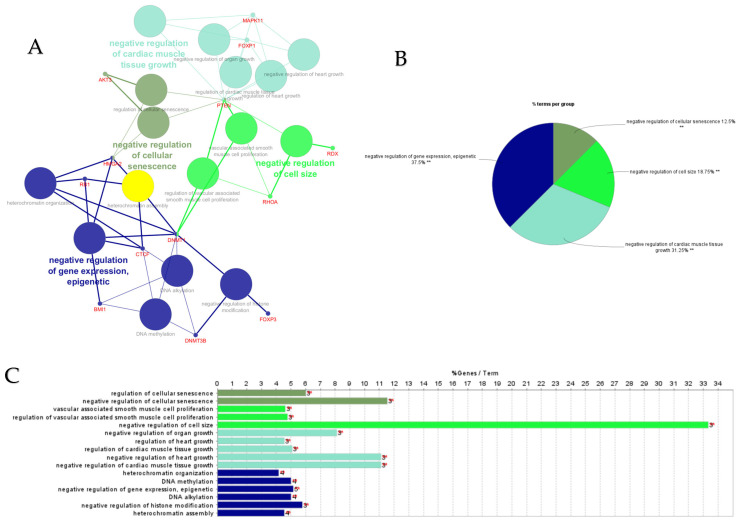
ClueGO analysis of downregulated hub genes in circulation in cows with metritis. (**A**) Functionally grouped network with terms as nodes linked based on their kappa score level (≥0.4), where only the label of the most significant term per group is shown. The node size represents the term enrichment significance. Functionally related groups partially overlap. The color gradient shows the gene proportion of each cluster associated with the term. (**B**) Overview chart with functional groups including specific terms for downregulated genes. The percentage of genes per term is shown as term label. ** *p* < 0.001. (**C**) GO/pathway terms specific for downregulated hub genes. The bars represent the number of genes associated with the terms. The number of genes per term is shown as bar label.

**Figure 8 cimb-47-00643-f008:**
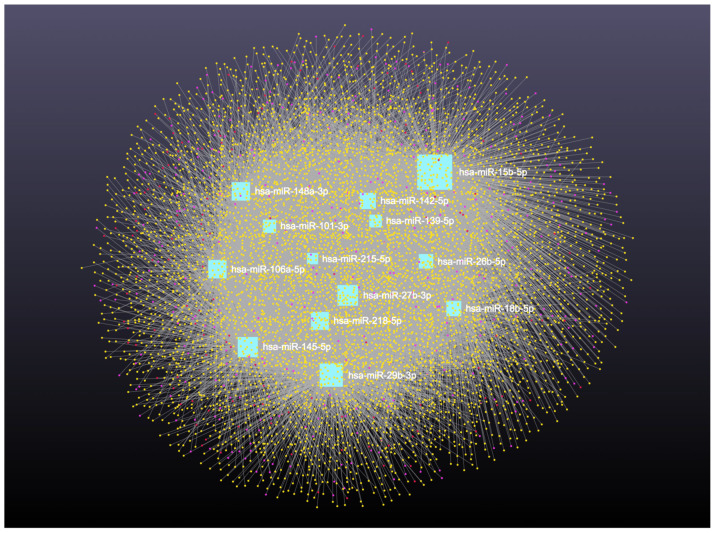
Upregulated circulating miRNAs in Holstein dairy cows with metritis. Network of interactions among upregulated miRNAs and their interacting circRNAs, lncRNAs, and snRNAs. Blue squares represent upregulated miRNAs, yellow circles represent circRNAs, purple circles represent lncRNAs, and orange circles represent scRNAs. A complete list of downregulated miRNAs, circRNAs, lncRNAs, and snRNAs, interactions are provided in Appendix A.

**Figure 9 cimb-47-00643-f009:**
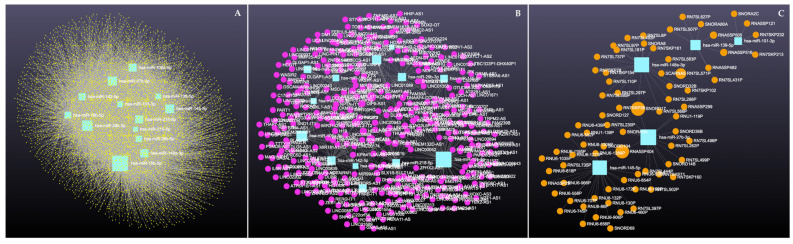
Circulating upregulated miRNAs in Holstein dairy cows with metritis. (**A**) Interaction network of upregulated miRNAs and circRNAs. Blue squares represent miRNAs and yellow circles represent circRNAs. (**B**) Interaction network of upregulated miRNAs and lncRNAs. Blue squares represent miRNAs, and purple circles represent lncRNAs. (**C**) Interaction network of upregulated miRNAs and snRNAs. Blue squares represent miRNAs and orange circles represent snRNAs. A complete list of upregulated miRNAs, circRNAs, lncRNAs, snRNAs, and mRNAs interactions are provided in Appendix A.

**Figure 10 cimb-47-00643-f010:**
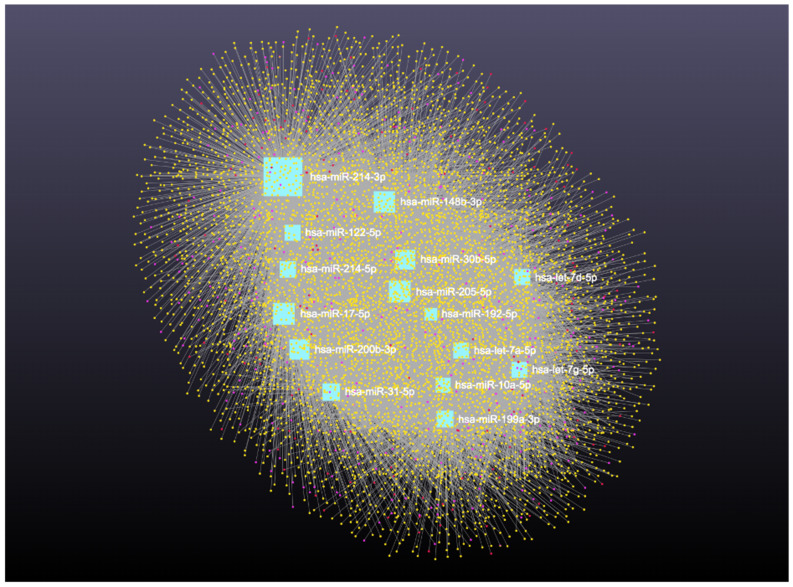
Downregulated circulating miRNAs in Holstein dairy cows with metritis. Network of interactions among downregulated miRNAs and their interacting circRNAs, lncRNAs, and snRNAs. Blue squares represent upregulated miRNAs, yellow circles represent circRNAs, purple circles represent lncRNAs, and orange circles represent snRNAs. A complete list of downregulated miRNAs, circRNAs, lncRNAs, and snRNAs, interactions are provided in Appendix A.

**Figure 11 cimb-47-00643-f011:**
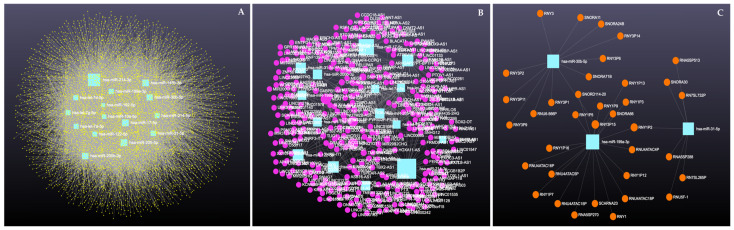
Circulating downregulated miRNAs in Holstein dairy cows with metritis. (**A**) Interaction network of downregulated miRNAs and circRNAs. Blue squares represent miRNAs and yellow circles represent circRNAs. (**B**) Interaction network of downregulated miRNAs and lncRNAs. Blue squares represent miRNAs, and purple circles represent lncRNAs. (**C**) Interaction network of downregulated miRNAs and snRNAs. Blue squares represent miRNAs and orange circles represent snRNAs. A complete list of downregulated miRNAs, circRNAs, lncRNAs, and snRNAs, interactions are provided in Appendix A.

**Table 1 cimb-47-00643-t001:** (**A**) Upregulated top 30 hub genes and their roles, human tissue expression, and protein–protein interactions (up to 6 closely related genes). (**B**) Downregulated top 30 hub genes and their roles, human tissue expressions, and protein–protein interactions (up to 6 closely related genes).

**(A)**			
**Top Hub Gene**	**Role**	**Human Tissue Expression**	**PPI**
AKT3	cell growth and division; cell survival; immune and inflammation response	ovary, uterus	MAP3K5, MAP3K5, FOXO3, PRKCZ, HSP90
MAML3	transcriptional coactivator for NOTCH proteins; regulation of retinoic acid	ovary, uterus	RA, NOTCH, IGF1R, AKT, IGF2, IL-1β
TGBR1	signal transduction; regulation of immune and inflammation responses	ovary, uterus, placenta	TGFB1, TGFBR2, ITGAV, LTBP1
TSHZ3	cell differentiation	ovary, uterus	APBB1
VEGFA	endothelial cell proliferation; promotes cell migration; inhibits apoptosis	ovary, uterus	FLT1, KDR, DLL4, HIF1A, MYOD1, STAT3, RUNX2, MYC,
CDK19	regulates RNA polymerase II and transcriptional activity; immune and inflammation responses	ovary, uterus	CCNC, MED3, MED13L, MED14, MED15
DDX6	cytoplasmic RNA regulation; negatively regulates immunity; proinflammation regulation	ovary, uterus	LSM14A, DCP1A, PATL1, CNOT1, EDC3, DDX6
LDLR	cell surface; cholesterol metabolism; promote inflammatory responses (Toll-Like Receptor, TLR)	ovary, uterus	DAB1, APOE, MYLIP, PCSK9, LRPAP1
MXD1	protein coding; transcriptional repressor; cell proliferation and differentiation; regulation of inflammation	ovary, uterus	AKT1, SUDS3, SIN3A, SIN3A, MAX
SOS2	intracellular Na^+^ and K^+^ homeostasis; cell maturation, cell survival and regulation of immune function	ovary, uterus	EGFR, KRAS, HRAS, GRB2, RASGRF1
USP9X	cell survival; immune response;	ovary, uterus, placenta	MCL1, CTNNB1, YAP1, UBE2D1, UNK
BRWD3	regulation of cell morphology and cytoskeletal organization	ovary, uterus	UBR1, UBXN7, CUL4A, CUL4B, WDR26
COL3A1	strengthen and support tissue; tissue repair	ovary, uterus, placenta	BMP1, PCOLCE, SPARC, DDR1, ERAL1
DNMT3A	DNA methylation	ovary, uterus, placenta	HISTIH3J, HISTIH3A, MPHOSPH8, H3F3B, DNMT3L
ELN	elasticity and tensile ability of tissue; tissue repair	ovary, uterus, placenta	LOX, LGALS3, FBLN1, FBLN2, FBN1
FBN1	cell formation; cell adhesion; fibrillogenesis; tissue repair	ovary, uterus, placenta	ELN, FBN1, FBN2, FBLN2, LTBP1, ZFP41
FBXW9	protein coding, MHC mediated	ovary, uterus	CDC6, SKP1, CUL1, UBA52, ELP2
TET1	immune response regulation	ovary, uterus	NANOG, OGT, ARID4B, SUDS3, SAP30
AFF4	protein coding	ovary, uterus, placenta	ELL, ELL2, CDK9, AFF4, CCNT1, MLLT1
GAB1	cellular growth; cell transformation; cellular apoptosis; regulation of immune (TLR) and inflammation responses	ovary, uterus, placenta	PTPN11, EGFR, PIK3R1, CRK, GRB2
PPARG	fatty acid storage and glucose metabolism; regulation of immune and inflammation responses	ovary, uterus, placenta	NCOA1, NCOR1, RXRA, PPRGC1A, MED1
ARFGEF1	cell–cell interactions; cell adhesion; cell migration and innate immunity	ovary, uterus, placenta	ARFGEF2, ARL1, SGPL1, AGPAT1, AGPAT2
CCNT2	protein coding	ovary, uterus, placenta	CDK9, CDK12, HEXIM1, AFF4, ELL3
DICER1	gene function regulation	ovary, uterus, placenta	AGO1, AGO2, AGO4, PRKRA, TARBP2
BBX	cell cycle progression	ovary, uterus, placenta	FOXJ2, SUMO2, HDAC1, SCGN, NIFK
CNOT2	cell proliferation; angiogenesis	ovary, uterus, placenta	CNOT1, RQCD1, CNOT3, CNOT4, CNOT7
CTGF	cell proliferation; angiogenesis; and tissue repair	ovary, uterus	FN1, ESR1, VEGFA, ITGA5, HSPG2, FBLN1
HIF1A	inflammation regulators	ovary, uterus, placenta	CREBBP, HIF1AN, ARNT, VHL, EP300, SUMO1
INADL	inflammation regulators	ovary, uterus, placenta	HOMER2, MPP5, LIN7B, AMOT, PLCB4
NCOA1	inflammatory and metabolic pathways	ovary, uterus, placenta	RARA, RXRA, PPARG, NR1I2, ESR1
**(B)**			
**Top Hub Genes**	**Roles**	**Human Tissue Expressions**	**PPIs**
UBE3A	ubiquitination; cell survival; immune response;	ovary, uterus, placenta	UBE2L3, TP53, PSMD4, UBE2D2, MCM7
ZC3H14	translation, mRNA stability; immune and inflammation response	ovary, uterus, placenta	SNIP1, RALYL, ARL61P4, LUC7L2, WRAP73
KDM5A	DNA methylation, apoptosis; immune response; inflammation (TLR)	ovary, uterus, placenta	RB1, HDAC1, TBP, RARA, GATAD1
CLIC4	regulation of cellular process; innate immune response; inflammation response;	ovary, uterus, placenta	CLIC2, CLIC5, CLIC6, TPRN, ESR2
CLIC5	regulation of cellular process; innate immune response; inflammation response;	ovary, uterus, placenta	CLIC2, CLIC4, EZR, FN1, TPRN
FKBP5	protein coding; promotes inflammation; immune response;	ovary, uterus, placenta	USP49, ESR1, PHLPP1, NR3C1, CDK9
LAMC1	regulates attachment, migration, and organization of cells; immune response; inflammation (TGFβ)	ovary, uterus, placenta	PYHIN1, NID2, LAMB1, NID1, SNAPIN
PIP4K2A	cell proliferation, differentiation, and motility; regulation of immune and inflammation response;	ovary, uterus, placenta	EPB41L3, CSNK2A2, ARL61P4, EAF1, ZRANB2
ZEB1	transcriptional repressor; activator of immune and inflammatory responses;	ovary, uterus	EP300, CTBP2, DRAP1, CTBP1, USP7
ZEB2	transcriptional repressor; activator of immune and inflammatory responses;	ovary, uterus, placenta	CTBP1, MTA1, MTA2, HDAC1, HDAC2
BDNF	cell proliferation and survival; regulation of immune and inflammation responses	ovary, uterus	NTRK2, NTF3, NTF4, SORCS2, SORT1
CLCC1	cell integrity; inflammation response	ovary, uterus, placenta	SPNS3, CUL3, HNRNPL, COX15, PTPRG, RNF4
SON	regulation of cell cycle; inflammation response	ovary, uterus, placenta	SRPK2, YWHAB, YWHAG, TRIP6, PRPF40A
ARID4A	DNA methylation; cell differentiation and proliferation	ovary, uterus, placenta	SAP30, BRMS1, ING2, RB1, HDAC1
ARID4B	DNA methylation; cell differentiation and proliferation	ovary, uterus, placenta	SIN3A, SAP130, HDAC1, HDAC2, ING2
EZH1	innate immune response; promotes TLR-triggered inflammatory cytokine production	ovary, uterus, placenta	SUZ2, EED, PHF1, PHF19, JARID2
FGD4	cell shape and integrity	ovary, uterus, placenta	ACTA1, GOLM1, FGD3, PRKCA, CAPZA2
RB1	regulates cell growth; immune and inflammatory response;	ovary, uterus, placenta	CDK2, CDK6, CEBPB, AHR, PPP1CA, RBBP9
DNA2	maintenance of mitochondrial and nuclear DNA stability; inflammation (ROS)	ovary, uterus	CIAO1, MMS19, RPA1, UPF1, FEN1
SMARCAD1	cell stability; mediates inflammation;	ovary, uterus, placenta	NUMA1, TRIM28, SUMO2, ZNF562, SUPT16H, DOK4
TTLL4	antiviral; immune and inflammation response	ovary, uterus	BRD7, NAE1, LZTS2, ATG16L1, KIAA12429
ZFYVE26	autophagy	ovary, uterus, placenta	PNMA5, TDO2, TFIP11, ALAS1, CEP44
PIK3CB	cell adhesion; immune (PIK3) and inflammation responses	ovary, uterus, placenta	PIK3R1, PIK3R2, PIK3R3, HCK, IRS1
EGLN3	hypoxia, immune and inflammation response	ovary, uterus	SIM1, SIM2, EPAS1, HIF1A, SIAH2
EZR	regulation of cytoskeleton and plasma membrane; immune response; proinflammation;	ovary, uterus, placenta	MSN, ARHGDIA, SLC9A3R1, SLC9A3R2, FADD
MLLT4	cell adhesion; innate immune response; chronic inflammation	ovary, uterus	PVRL1, PVRL2, PVRL3, SSX2IP, HRAS
PLCB1	mediates intercellular signaling; regulation of immune response; vascular inflammation	ovary, uterus, placenta	SLC9A3R1, KRAS, GNAQ, TRPC3, GNA11

**Table 2 cimb-47-00643-t002:** Interferon-stimulated genes and associated up- and downregulated miRNAs.

Interferon Stimulated Genes	Upregulated miRNAs	Downregulated miRNAs
ISG15	miR-27b;	miR-148b;
RSAD2	miR-26b; miR-27b;	miR-17;
CTSL	miR-15b;	miR-200b; let-7d;
CXCL family	CXCL2; CXCL5; miR-15b; CXCL6; CXCL9; CXCL13; miR-26b; CXCL1; CXCL2; CXCL10; CXCL11; CXCL12; CXCL16; miR-27b; CXCL1; CXCL3; CXCL8; miR-101-3p; CXCL6; CXCL8; miR-106a; CXCL10; miR-139; CXCL3; miR-142; CXCL2; miR-215;	CXCL1; CXCL16; miR-205; CXCL1; miR-214; CXCL2; miR-192; CXCL6; CXCL11; miR30b; CXCL10; CXCL12; miR-31; CXCL1; CXCL2; CXCL5; CXCL8; miR-148b; CXCL8; miR-17;
solute-carrier (SLC) gene family	miR-26b; miR-27b; miR-29b; miR-101; miR-106a; miR-139; miR-142; miR-145; miR-148; miR-15b; miR-17; miR-18b; miR-215; miR-218; miR-22;	miR-200b; miR-205; miR-214; miR-30b; miR-31; let-7a; let-7d; let-7g; miR-10a; miR-122; miR-148b; miR-17; miR-192; miR-199a;

**Table 3 cimb-47-00643-t003:** Top 5 coding and noncoding RNAs with high degree and betweenness centrality targeted by up- and downregulated miRNAs, their endometrial expressions, and potential roles.

ID	Degree †	Betweenness ‡	Endometrial Protein Expression *	Potential Role in Uterine Disease
** *Upregulated lncRNA* **				
KCNQ1OT1	11	4996.728	4.7	dysregulation could potentially affect the expression of genes involved in immune responses and tissue repair
NEAT1	11	4392.385	34.6	increased expression during uterine infections may contribute to chronic inflammation and impaired tissue repair, leading to subfertility
XIST	10	3645.005	-	aberrant expression could influence the expression of genes involved in immune regulation and tissue remodeling in the uterus
HELLPAR	7	2116.966	-	ubiquitination and cell cycle progression; upregulation during uterine infections may exacerbate inflammation and disrupt normal uterine function, contributing to fertility issues
TUG1	7	2007.66	58.7	altered TUG1 expression could affect epithelial cell function and immune responses, potentially leading to uterine disease and subfertility
** *Upregulated circRNA* **				
NBPF9	13	22,830.78	6.1	not well-established, but dysregulation could potentially affect gene expression pathways related to immune responses and tissue repair
NBPF10	13	22,830.78	4.9	not well-established, but dysregulation could potentially affect gene expression pathways related to immune responses and tissue repair
RANBP2	13	22,830.78	130.5	altered expression could disrupt the transport of proteins and RNAs essential for uterine cell function, contributing to disease progression
RGPD4	13	22,830.78	0.2	upregulation during uterine infections may affect signaling pathways crucial for immune responses and tissue remodeling, leading to persistent inflammation and fertility issues
RGPD6	13	22,830.78	0.0	upregulation during uterine infections may affect signaling pathways crucial for immune responses and tissue remodeling, leading to persistent inflammation and fertility issues
** *Upregulated sncRNA* **				
SNORD17	2	1518	-	upregulation during uterine disease could indicate an attempt to correct RNA modifications, but persistent changes may reflect ongoing cellular stress and dysfunction
RNA5SP404	2	1450	-	altered expression during uterine infections may disrupt protein synthesis in uterine cells, impairing their function and contributing to disease progression
RN7SL571P	2	858	-	upregulation during uterine disease could indicate an attempt to enhance protein targeting to the endoplasmic reticulum, but persistent alterations may disrupt normal cellular processes
RNA5SP505	2	182.5	-	altered expression during uterine infections may disrupt protein synthesis in uterine cells, impairing their function and contributing to disease progression
RNA5SP516	2	182.5	-	altered expression during uterine infections may disrupt protein synthesis in uterine cells, impairing their function and contributing to disease progression
** *Downregulated lncRNA* **				
KCNQ1OT1	14	6142.142	4.7	downregulation may lead to loss of imprinting and dysregulated gene expression, potentially affecting uterine cell function and immune responses
NEAT1	14	6084.598	34.6	downregulation of NEAT1 has been associated with impaired immune responses and tissue repair; in dairy cows, decreased NEAT1 expression during uterine infections may contribute to chronic inflammation and impaired tissue repair, leading to subfertility
XIST	13	5449.031	-	aberrant expression of XIST could influence the expression of genes involved in immune regulation and tissue remodeling in the uterus; downregulation may disrupt these processes, contributing to uterine disease progression
SNHG16	9	2011.856	-	implicated in various biological processes, including cell proliferation and differentiation; altered expression could affect epithelial cell function and immune responses in the uterus, potentially leading to uterine disease and subfertility
HCG18	8	1573.84	-	downregulation of may impair trophoblast invasion and placental development, leading to uterine dysfunction and fertility issues
** *Downregulated circRNA* **				
NBPF9	14	30,638.8	6.1	dysregulation could potentially affect gene expression pathways related to immune responses and tissue repair
UBR4	11	30,638.8	23.8	altered expression could disrupt the degradation of proteins essential for uterine cell function, contributing to disease progression
KPNA6	10	30,638.8	41.6	downregulation could disrupt the transport of proteins and RNAs essential for uterine cell function, contributing to disease progression
MACF1	10	30,638.8	154	altered expression could affect the structural integrity of uterine cells, impairing tissue repair and immune responses
USP24	10	30,638.8	12.6	downregulation could disrupt the regulation of protein degradation pathways, affecting uterine cell function and contributing to disease progression
** *Downregulated sncRNA* **				
RNY3P15	2	312	-	downregulation may impair RNA processing and affect the expression of genes involved in immune responses and tissue repair in the uterus
RNY1P2	2	217	-	altered expression could affect the processing of precursor RNAs, leading to dysregulated gene expression and contributing to uterine disease
RNU4ATAC4P	2	183	-	downregulation may impair RNA splicing, affecting the expression of genes involved in immune responses and tissue repair in the uterus
SNORA66	2	183	-	downregulation during uterine disease could indicate an attempt to correct RNA modifications, but persistent changes may reflect ongoing cellular stress and dysfunction
RNY1P13	2	172	-	altered expression could affect the processing of precursor RNAs, leading to dysregulated gene expression and contributing to uterine disease

† Degree centrality is a measure of network centrality that quantifies the number of direct connections a node has to other nodes in a network. ‡ Betweenness centrality is a measure of the number of times a given node lies on one of the paths between all pairs of nodes in the network. * nTPM, 5085 Glandular and luminal cells.

## Data Availability

The data presented in this study are available in the article or Appendix A herein.

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
