# Peer review of "Integrative Analysis of Differentially Expressed miRNAs and Noncoding RNA Networks Reveals Molecular Mechanisms Underlying Metritis in Postpartum Dairy Cows"

_cimb, 2025, doi:10.3390/cimb47080643_

Round 1
Reviewer 1 Report
Comments and Suggestions for Authors
Review on the manuscript titled “Integrative Analysis of Differentially Expressed miRNAs and Non-Coding RNA Networks Reveals Molecular Mechanisms Underlying Metritis in Postpartum Dairy Cows” by Kasimanickam et al., 2025.
The authors addressed the issue of miRNA and ncRNA underlying Metritis impact in Dairy Cows.
In comprehensive introduction the authors assessed postpartum uterine Metritis diseases spectra and its impact on reproductive performance and overall health. The authors correspondingly assessed immune response genes network including cytokines and other genes, along with the impact of miRNA and ncRNA along the way.
Also the authors note that “In a previous study, 84 circulating miRNAs were profiled in postpartum dairy cows, identifying 30 differentially expressed miRNAs—16 upregulated and 14 downregulated—in cows with metritis compared to healthy controls”.
The authors ultimate goal was focusing on these RNAs within the Endometriosis/metritis issues.
- Materials and Methods contains several chapters.
Data: The authors rendered Their previous studies data ([18], Holstein dairy cows with metritis (n = 4) and healthy controls (n = 4)
2.3 Conserved nucleotide sequences
Nucleotide sequences of DE-miRNAs were retrieved from miRBase, (www.mirbase.org accessed on 10 January 2025) and compared for similarities among human and cattle [19,20].
2.4 Prediction and analysis of target genes of differentially expressed miRNAs
2.5 Construction of protein-protein interaction network and screening of hub gene
2.6 Gene ontology and functional annotation analysis
2.7 miRNA, circRNA, lncRNA, snRNA, and mRNA Interaction Network
- Results section comprises several chapters:
3.1 miRNA, gene and protein-protein interactions
3.2 Interaction Network Among miRNA, circRNA, lncRNA, snRNA, and mRNA
3.2. Figures, Tables and Schemes – this chapter title is irrelevant, should be removed.
Chapter 3.2. comprises 3 tables (Table1A,B, Table2, Table 3), titled:
Table 1A. Up-regulated top 30 hub genes, and their roles, human tissue expression and protein-protein interactions (up to 6 closely related genes)
Table 1B. Down-regulated top 30 hub genes, and their roles, human tissue expressions and protein-protein interactions (up to 6 closely related genes)
Table 2. Interferon-stimulated genes and associated up- and down-regulated miRNAs in cows with metritis.
Table 3. The top 5 coding and non-coding RNAs with high degree and betweenness centrality targeted by up- and down-regulated miRNAs, their endometrial expressions and potential roles
After the tables immediately follow 5 figures:
Figure 1. Fold regulation of differentially expressed miRNAs in Holstein cows with metritis
Figure 2. STRING protein-protein interaction (PPI) network
Figure 3. ClueGO analysis of up-regulated genes in circulation in cows with metritis.
Figure 4. ClueGO analysis of down-regulated genes in circulation in cows with metritis.
Figure 5. Protein-Protein Interaction (PPI) network of hub genes of DE-miRNAs
Figure 6. ClueGO analysis of up-regulated hub genes in circulation in cows with metritis.
Figure 7. ClueGO analysis of down-regulated hub genes in circulation in cows with metritis
Figure 8. Upregulated circulating miRNAs in Holstein dairy cows with metritis.
Figure 9. Circulating upregulated miRNAs in Holstein dairy cows with metritis.
Figure 10. Downregulated circulating miRNAs in Holstein dairy cows with metritis.
Figure 11. Circulating downregulated miRNAs in Holstein dairy cows with metritis.
After the comprehensive discussion section, the authors make some conclusions on the study including: “The findings suggest that miRNA-mediated regulation—alongside its interaction with other non-coding RNAs and metabolic pathways—plays a central role in the development of uterine diseases.
These insights offer potential biomarkers and therapeutic targets for improving uterine recovery, enhancing fertility, and reducing the burden of reproductive disorders in dairy herds.”
The supplemental info comprises 7 supplementary tables containing elaboration on the miRNA expression rates and other relevant information.
Overall, the manuscript is a follow-up on the study the authors published in 2016 (ref 18). The manuscript would be of interest to the researchers in the field. Still, while the authors performed a great deal of work, the structure and presentation of the results should be addressed. Below are the notes.
- S71: Data from a previous study [18] were used-> Data from OUR previous study [18]
- “Qiagen miScript PCR array targeting 84 bovine miRs” – while the authors use the previous data in the manuscript, it’s unclear whether they’ve sequenced it repeatedly, or just took from the previous study. Please, elaborate/reference accordingly.
- 3 Conserved nucleotide sequences – this title is poorly informative, please, expand.
- It’s unclear why authors use human ontologies instead of bovine ones in tables 1, 2. Please, elaborate.
- Table 3: “Betweenness” column, what’s that? Is it the distance between aim and ncRNA target? Please, explain the columns explicitly in the Table’s header, and use more appropriate terms.
- “Despite(s) its original role(s) in X-chromosome dosage compensation, lncRNA Xist also participates in the regulatory network of lncRNA Xist participated in various signaling pathways, such as TGF-beta signaling pathway, PIK3/AKT signaling pathway, Wnt/β-catenin signaling pathway, FOXO signaling pathway, NF-kB signaling pathway, mTOR signaling pathway, MAPK signaling pathway, Toll-like receptor signaling pathway, JAK-STAT signaling pathway, T cell receptor signaling pathway, and B cell receptor signaling pathway may therefore plausibly play a role in pathophysiology of uterine diseases in dairy cows [86,87].” – please, check up the punctuation and grammar in this sentence. Also, the studies mentioned herein [86, 87] relate to 3D chromatin conformation, not the particular pathways. Thus, this passage should either be removed, or explicitly elaborated on.
- Overall, there are some typos along the whole manuscript, please, address.
- The statement “The findings suggest that miRNA-mediated regulation—alongside its interaction with other non-coding RNAs and metabolic pathways—plays a central role in the development of uterine diseases” is overrated, to my opinion. Please, use more moderate statement.
- I cannot see any references to figures in the text.
- Figures 8-11 are incomprehensive in their current form, please use more comprehensive/clear design, or remove them.
- Figures should be referenced within the text and placed accordingly.
Comments on the Quality of English Language
There are some typos in the manuscript, should be spell checked.
Author Response
We sincerely thank the reviewer for the constructive and insightful comments. We have carefully considered each point and made the necessary revisions to address them appropriately in the revised manuscript.
Reviewer Comment:
S71: “Data from a previous study [18] were used” → “Data from OUR previous study [18]”
Authors: We appreciate the suggestion. The phrase has been updated throughout the manuscript to reflect “our previous study [18]” for clarity and consistency.
Reviewer Comment:
“Qiagen miScript PCR array targeting 84 bovine miRs” – while the authors use the previous data in the manuscript, it’s unclear whether they’ve sequenced it repeatedly or just reused the previous data. Please elaborate/reference accordingly.
Authors: Thank you for pointing this out. We have clarified in the Methods section that the data used were obtained from our previously published study and were not re-sequenced for this analysis.
Reviewer Comment:
Section title “3 Conserved nucleotide sequences” is poorly informative—please expand.
Authors: The section title has been revised for better clarity and informativeness. Please refer to Lines 87–89 in the revised manuscript for the updated context.
Reviewer Comment:
It’s unclear why the authors use human ontologies instead of bovine ones in Tables 1 and 2. Please elaborate.
Authors: Thank you for your observation. We have clarified in the revised manuscript that human ontologies were used due to the more comprehensive annotation available, which enables more detailed functional insights. This has been explained in the Methods section.
Reviewer Comment:
Table 3: “Betweenness” column—what does this refer to? Is it the distance between the ncRNA and its target? Please explain column headers explicitly and use appropriate terms.
Authors: We appreciate the comment. We have clarified the meaning of the “Betweenness” column in the table’s footnote. It refers to a centrality measure used in network analysis to indicate the importance of a node in facilitating communication within the network.
Reviewer Comment:
The sentence about lncRNA Xist is grammatically incorrect and confusing. This passage should be removed or revised.
Authors: Thank you for highlighting this. The sentence has been reformatted for clarity and grammatical accuracy. We have also revised the content to better align with the scope of references [86, 87], and we have removed any unsupported pathway associations.
Reviewer Comment:
There are typos throughout the manuscript. Please address them.
Authors: We have thoroughly proofread the manuscript and corrected typographical and grammatical errors to improve readability and overall language quality.
Reviewer Comment:
The statement “The findings suggest that miRNA-mediated regulation—alongside its interaction with other non-coding RNAs and metabolic pathways—plays a central role in the development of uterine diseases” is overstated. Please moderate the language.
Authors: We agree and have rephrased the sentence to present the findings in a more cautious and evidence-based manner.
Reviewer Comment:
There are no references to figures in the main text.
Authors: All relevant references to figures have now been included in the Results section to improve the flow and contextual understanding of the data presented.
Reviewer Comment:
Figures 8–11 are currently incomprehensible. Please redesign for clarity or consider removing them.
Authors: We acknowledge the complexity of Figures 8–11. To enhance clarity, we have provided the original high-resolution versions as supplementary files. A reference to these supplementary figures is included in the main text, directing readers for more detailed exploration of miRNAs and ncRNAs.
Reviewer 2 Report
Comments and Suggestions for Authors
The present manuscript analyses the expressed miRNA and non-coding RNA networks comparing healthy and metritis cows.
The content of the present manuscript is very interesting and sheds light on the molecular mechanisms of postpartum metritis in cows. From my point of view, the manuscript is worth publishing. However, it needs a language arrangement. Some sentences are hard to understand, and grammar mistakes have been detected throughout the text. Please submit the manuscript to a language reviewer before resubmitting it to the journal.
Something else that needs to be addressed is the fact that the authors do not mention any relevant protein in the text. They name them in the abstract, but in the results chapter, they only state that the results are presented in this or that table. Please name the more relevant proteins in the text.
A full-page conclusion? That is too much. Please, shorten it.
Comments on the Quality of English LanguageLanguage MUST be improved
Author Response
Reviewer Comment:
The content of the present manuscript is very interesting and sheds light on the molecular mechanisms of postpartum metritis in cows. From my point of view, the manuscript is worth publishing. However, it needs a language arrangement. Some sentences are hard to understand, and grammar mistakes have been detected throughout the text. Please submit the manuscript to a language reviewer before resubmitting it to the journal.
Author Response:
We appreciate the reviewer’s positive assessment of the manuscript. The entire manuscript has been carefully reviewed and revised by experts proficient in English to improve clarity, grammar, and overall language quality.
Reviewer Comment:
Something else that needs to be addressed is the fact that the authors do not mention any relevant protein in the text. They name them in the abstract, but in the results chapter, they only state that the results are presented in this or that table. Please name the more relevant proteins in the text.
Author Response:
Thank you for your observation. To avoid redundancy and maintain readability, we referred readers to the detailed data presented in the tables. However, we have highlighted the most relevant proteins, while maintaining comprehensive analysis, in the Discussion.
Reviewer Comment:
A full-page conclusion? That is too much. Please, shorten it.
Author Response:
We aimed to provide a comprehensive summary of our findings and their implications in the Conclusion section. While there is no specific word limit for this section, we understand the concern and have revised and shortened the Conclusion to improve conciseness while preserving the key messages.
Reviewer 3 Report
Comments and Suggestions for Authors
This study focuses on the molecular mechanisms of postpartum metritis in dairy cows. They have revised the manuscript and significantly improved. I suggest to accept and publish in your journal.
Author Response
Comment: This study focuses on the molecular mechanisms of postpartum metritis in dairy cows. They have revised the manuscript and significantly improved. I suggest to accept and publish in your journal.
Authors: We sincerely thank the reviewer for their time and positive evaluation of our revised manuscript. We are grateful for your constructive feedback throughout the review process and are pleased to know that you found the manuscript significantly improved. We appreciate your recommendation for acceptance and publication.
Round 2
Reviewer 1 Report
Comments and Suggestions for Authors
The authors have adequately addressed my comments, but there are minor edition notes:
Fig. 7 plots are not assigned by (A, B, C) in the fig 7, please, supply.
If there is possibility to enlarge the font of GO titles in Fig.7C, please do, otherwise it would be appropriate to provide GO annotations in supplement and referencing it in the caption as it's done in Fig. 10.
Fig. 8 caption is truncated: "purple circles represent lncRNAs, and orange (circles represent snRNAs)? ", also reference would be relevant as in Fig. 10.
Please note that Fig. 9 is absent, you probably need to re-numerate the figures accordingly.
Author Response
We would like to thank the reviewer for the additional comments. Below, we provide point-by-point responses.
All images resolution were increased to 1200 pixel.
Comment: Fig. 7 plots are not labeled as (A, B, C). Please add these labels.
Authors: The labels (A, B, C) have been added to Figure 7.
Comment: If possible, enlarge the font size of the GO titles in Fig. 7C. If not, consider including GO annotations in the supplementary material and referencing them in the caption, as done in Fig. 10.
Authors: The image resolution has been increased to 1200 pixels to enhance readability.
Comment: The caption for Fig. 8 is truncated: "purple circles represent lncRNAs, and orange (circles represent snRNAs)?". A reference similar to the one in Fig. 10 would also be helpful.
Authors: The complete caption has been added to Figure 8.
Comment: Please note that Fig. 9 is missing. You may need to re-number the figures accordingly.
Authors: Figure 9 has been included.